# Development of RNA/DNA Hydrogel Targeting Toll-Like Receptor 7/8 for Sustained RNA Release and Potent Immune Activation

**DOI:** 10.3390/molecules25030728

**Published:** 2020-02-07

**Authors:** Fusae Komura, Kana Okuzumi, Yuki Takahashi, Yoshinobu Takakura, Makiya Nishikawa

**Affiliations:** 1Department of Biopharmaceutics and Drug Metabolism, Graduate School of Pharmaceutical Sciences, Kyoto University, Sakyo-ku, Kyoto 606-8501, Japan; fusae.komura@gmail.com (F.K.); okuzumi.kana.27z@st.kyoto-u.ac.jp (K.O.); ytakahashi@pharm.kyoto-u.ac.jp (Y.T.); takakura@pharm.kyoto-u.ac.jp (Y.T.); 2Faculty of Pharmaceutical Sciences, Tokyo University of Science, Noda, Chiba 278-8510, Japan

**Keywords:** RNA, adjuvant, Toll-like receptor 7/8, hydrogel, DNA nanotechnology

## Abstract

Guanosine- and uridine-rich single-stranded RNA (GU-rich RNA) is an agonist of Toll-like receptor (TLR) 7 and TLR8 and induces strong immune responses. A nanostructured GU-rich RNA/DNA assembly prepared using DNA nanotechnology can be used as an adjuvant capable of improving the biological stability of RNA and promoting efficient RNA delivery to target immune cells. To achieve a sustained supply of GU-rich RNA to immune cells, we developed a GU-rich RNA/DNA hydrogel (RDgel) using nanostructured GU-rich RNA/DNA assembly, from which GU-rich RNA can be released in a sustained manner. A hexapod-like GU-rich RNA/DNA nanostructure, or hexapodRD6, was designed using a 20-mer phosphorothioate-stabilized GU-rich RNA and six phosphodiester DNAs. Two sets of hexapodRD6 were mixed to obtain RDgel. Under serum-containing conditions, GU-rich RNA was gradually released from the RDgel. Fluorescently labeled GU-rich RNA was efficiently taken up by DC2.4 murine dendritic cells and induced a high level of tumor necrosis factor-α release from these cells when it was incorporated into RDgel. These results indicate that the RDgel constructed using DNA nanotechnology can be a useful adjuvant in cancer therapy with sustained RNA release and high immunostimulatory activity.

## 1. Introduction

Guanosine- and uridine-rich single-stranded RNA (GU-rich RNA) is a typical agonist of Toll-like receptor 7 (TLR7) and TLR8 in the endosomes of most antigen presenting cells [1,2,3,4,5]. Ligation of GU-rich RNA to TLR7/8 induces the release of several types of cytokines. TLR7 and TLR8 are expressed abundantly in antigen presenting cells, especially in dendritic cells and monocytes that are important to trigger antigen-specific immune responses. Therefore, GU-rich RNA is expected to be a promising immune adjuvant.

In the development of a GU-rich RNA-based adjuvant, efficient delivery to endosomes of antigen presenting cells expressing TLR7/8 is necessary. However, endocytic RNA uptake by antigen presenting cells is limited and single-stranded RNA is easily hydrolyzed by the RNase present in biological environments [6,7,8]. Moreover, RNA is rapidly cleared from the administration site. These problems can be circumvented by complex formation with lipid nanoparticles, but the complicated manufacture of lipid nanoparticles and their high manufacturing costs are major hurdles for their clinical applications [9]. Moreover, various types of sustained release systems, such as microparticles and nanofibers, for bioactive RNAs have been reported [10,11,12]. Hydrogels constitute one such system. Hydrogels consisting of various polymers, such as polyethylenimine, have been reported. However, the hydrogel systems developed so far have some problems, including complicated preparation processes. An approach that can solve these problems is thus desired.

As an approach to solve these problems, we developed a tetrapod-like nanostructured GU-rich RNA/DNA assembly, or tetrapodRD3, comprising one phosphorothioate-stabilized RNA and three phosphodiester DNAs [13]. The tetrapodRD3 showed improved biological stability, efficient uptake by antigen presenting cells, high immunostimulatory activity, and high antigen presentation. The potency of such GU-rich RNA-based adjuvants can be further increased by their sustained release. Nucleic acid-based hydrogels, such as DNA hydrogels [14], can be a promising sustained release system for GU-rich RNA compared to conventional systems, because they can be simply prepared by annealing and mixing their components.

Based on these considerations, this study aimed to develop a new RNA adjuvant capable of sustained RNA release, targeting TLR7/8. To this end, we designed a hexapod-like nanostructured GU-rich RNA/DNA assembly, or hexapodRD6, consisting of a 20-mer phosphorothioate GU-rich RNA and six phosphodiester DNAs. Two sets of hexapodRD6, i.e., hexapodRD6-1 and hexapodRD6-2, both of which had complementary cohesive ends to each other, were prepared and mixed together to obtain the GU-rich RNA/DNA hydrogel, or RDgel. We first evaluated RDgel formation and GU-rich RNA release from the hydrogel. We then examined the cellular uptake of the RDgel and tumor necrosis factor (TNF)-α release after addition to mouse macrophage-like RAW264.7 cells and mouse dendritic DC2.4 cells. Finally, we added the conditioned media of RDgel-treated RAW264.7 cells to mouse colon26 tumor cells and examined their effects on tumor cell proliferation.

## 2. Results

### 2.1. Preparation of RDgel

Figure 1a shows the schematic images of hexapodRD6-1 and hexapodRD6-2. HexapodRD6-1 consists of oligodeoxyribonucleotide (ODN)-1, ODN-2, ODN-3, ODN-4, ODN-5, ODN-6, and oligoribonucleotide (ORN)-1 (GU-rich RNA), and hexapodRD6-2 consists of ODN-7, ODN-8, ODN-9, ODN-10, ODN-11, ODN-12, and ORN-1. Three of the six pods of hexapodRD6-1 and hexapodRD6-2 had a 20-base sequence complementary to ORN-1. An 8-base cohesive sequence complementary to another hexapodRD6 was attached at the end of the other three pods. GU-rich RNA/DNA hydrogel, or RDgel, was obtained by mixing hexapodRD6-1 and hexapodRD6-2 at a 1:1 molar ratio. HexapodD6-1 (ODN-1, ODN-2, ODN-3, ODN-4, ODN-5, and ODN-6) and hexapodD6-2 (ODN-7, ODN-8, ODN-9, ODN-10, ODN-11, and ODN-12) were also designed to confirm the loading of ORN-1 onto hexapodRD6. Table 1 shows the oligonucleotide sequences used for the RDgel.

Formation of hexapodRD6-1, hexapodRD6-2, and RDgel was evaluated by polyacrylamide gel electrophoresis (PAGE) analyses (Figure 1b,c). HexapodRD6-1 and hexapodRD6-2 majorly appeared as a single band at a position different from that of the ORN-1 band. In addition, the bands of hexapodRD6-1 and hexapodRD6-2 were different from the bands of hexapodD6-1 and hexapodD6-2, indicating that ORN-1 was loaded onto hexapodRD6-1 and hexapodRD6-2. The RDgel hardly migrated into the gel.

Hydrogel formation was also evaluated by adding PBS onto the PI-stained hexapodRD6-1 or RDgel. HexapodRD6-1 was rapidly mixed with PBS and was diluted (Figure 1d). On the other hand, RDgel remained separate from PBS (Figure 1e). These results indicate that simple mixing of hexapodRD6-1 and hexapodRD6-2 resulted in hydrogel formation.

### 2.2. RNA Release from RDgel

The release profile of GU-rich RNA from RDgel was evaluated using a Transwell chamber system, in which FAM-ORN-1 released from RDgel could be detected in the lower well. Figure 2a shows the time course of the amount of RNA/DNA remaining in the chamber with 0.4 μm pore. No significant amount of FAM-hexapodRD6-1 remained on the chamber at 4 h after incubation, whereas FAM-RDgel remained on the Transwell chamber. In addition, FAM-ORN-1 was more slowly released from RDgel than from hexapodRD6 (Figure 2b). These results indicate that RDgel was gradually decomposed and that GU-rich single-stranded RNA was slowly released from the hydrogel. 

### 2.3. Optimization of Preparation Conditions of RDgel

Release of ORN-1 from RDgel can be prolonged by controlling the RDgel preparation conditions. Two parameters, the concentrations of nucleotides and metal ions, were selected and RDgel was prepared under various conditions. Figure 3 shows the time courses of FAM-ORN-1 release from RDgel prepared with different nucleotide concentrations. Increasing the nucleotide concentration increased the amount of RDgel containing FAM-ORN-1 on the Transwell chamber. In addition, the release of FAM-ORN-1 from the RDgel was delayed.

Figure 4 shows the FAM-ORN-1 released from FAM-RNA/DNA hydrogel prepared using different metal ions. Sodium ions and magnesium ions, which are also present in the body, were selected. Increasing the metal ion concentration increased the amount of RDgel on the Transwell chamber, and delayed the release of FAM-ORN-1. The release profiles of FAM-ORN-1 from RDgel were comparable between the one prepared with 150 mM sodium ions and that with 50 mM magnesium ions. Considering the concentrations of metal ions in blood and extracellular fluids, RDgel prepared with 150 mM sodium ions was used for the following experiments.

### 2.4. RNA Release from RDgel under Serum-Containing Conditions

Figure 5 shows the time course of FAM-ORN-1 released from RDgel under 10% serum-containing conditions. Fetal bovine serum (FBS) was used as the serum. FBS-containing solution was overlaid onto FAM-RDgel and FAM-ORN-1 released into the solution was measured over time. The presence of FBS hardly affected the FAM-ORN-1 released from RDgel. This result indicates that GU-rich single-stranded RNA (ssRNA) can be gradually released from RDgel under serum-containing conditions.

### 2.5. Cellular Uptake of RDgel by Dendritic Cells

RDgel uptake by mouse dendritic DC2.4 cells was evaluated by flow cytometry (Figure 6). The mean fluorescence intensity (MFI) of DC2.4 cells after addition of RDgel containing FAM-ORN-1 was found to increase with time. The MFI of DC2.4 cells after addition of RDgel containing FAM-ORN-1 was higher than after addition of hexapodRD6 containing FAM-ORN-1.

### 2.6. TNF-α Release after Addition of RDgel to Immune Cells

The immunostimulatory activity of RDgel was evaluated by the TNF-α release from DC2.4 cells (Figure 7a). DC2.4 cells showed significantly higher TNF-α release after RDgel addition than after addition of ORN-1 or hexapodRD6. Similar results were obtained when mouse macrophage-like RAW264.7 cells were used (Figure 7b). These results indicate that hydrogel formation significantly increased the immunostimulatory activity of ORN-1.

### 2.7. Effects of the Conditioned Media of RDgel-Treated RAW264.7 Cells on Colon26 Tumor Cell Proliferation

ORN-1, hexapodRD6-1, or RDgel was added to RAW264.7 cells, and the culture supernatant was collected as conditioned medium at 24 h after addition. The conditioned medium was then added to mouse colon26 tumor cells. Figure 8a shows the number of colon26 cells at 24 h after adding the conditioned media. The conditioned media of RDgel-treated RAW264.7 cells significantly reduced the number of colon26 cells. The apoptotic cells were then detected by staining the cells with annexin V, an early apoptosis marker, and propidium iodide (PI), a late apoptosis marker. In the RDgel group, the proportion of annexin V and PI double-positive cells was increased compared with that in the ORN-1 and hexapodRD6 groups (Figure 8b). These results suggest that immunostimulatory factors such as cytokines released from RDgel-treated RAW264.7 cells induced cell death including apoptosis in colon26 cells.

## 3. Discussion

GU-rich ssRNA is an adjuvant that can efficiently induce immune responses and increase vaccine potency. To enhance the effectiveness of vaccines, sustained stimulation of the innate immune system by an adjuvant is desirable. Although hydrogels can be used for sustained RNA release, conventional hydrogels have various problems such as complicated preparation processes. The newly developed RDgel in the present study was composed only of nucleic acids and was easy to prepare. This study revealed an approach that can solve the problems associated with conventional sustained RNA release systems.

The release rate of FAM-ORN-1 from FAM-RDgel was faster than the disintegration rate of FAM-RDgel. There are mainly three types of RNA forms released from RDgel: ORN-1, hexapodRD6, and the connecting structure of multiple hexapodRD6. RDgel is disintegrated and oligonucleotides and/or nanostructured RNA are then released from RDgel. ORN-1 conjugated to the end of hexapodRD6 can be dissociated without hydrogel disintegration. Therefore, it is considered that the RNA release rate calculated based on the fluorescence intensity derived from FAM was higher than the gel disintegration rate. The high immunostimulatory activity induced by RDgel is considered a result of efficient cellular uptake of RDgel. It was expected that the connecting structure of multiple hexapodRD6 released from RDgel results in a high cellular uptake of RNA. It was assumed that RDgel was not taken up by the cell as it was, but that RNA, hexapodRD6, or the connecting structure of multiple hexapodRD6 released from the RDgel was taken up by DC2.4 cells. Polypod-like nanostructured DNA (polypodna) are recognized by the nucleic acid receptors expressed on the surface of immune cells and are delivered to endosomes by clathrin-mediated endocytosis. We previously reported that polypodna are taken up, at least partially, through macrophage scavenger receptor-1 (MSR1) [15]. MSR1 is a receptor for negatively charged particles such as low density lipoprotein [16], and it efficiently recognizes the connecting structure of multiple hexapodRD6 larger than hexapodRD6.

The time course of FAM-ORN-1 from RDgel was delayed in proportion to the concentration of RNA/DNA and metal ions (Figure 4). It is presumed that the increase in the nucleotide concentration of RDgel increased the density of the network structure formed by the binding of hexapodRD6, and the increased concentration of metal ions promoted stabilization of the double-stranded structure.

We previously reported that GU-rich ORN containing the HIV-1 U5 region and tetrapod-like RNA/DNA nanostructure with GU-rich ORN were specifically recognized by human TLR8 [13]. Therefore, it is assumed that hexapodRD6 and RDgel were recognized by human TLR8 in the same manner as tetrapodRD3.

TLR7/8 activation is reported to induce an anti-tumor effect by the release of cytokines such as TNF-α and interleukin-12 [17,18] and by activation of cytotoxic T lymphocytes [19,20,21]. The conditioned medium of immune cells treated with RDgel contained TNF-α and other cytokines induced by RDgel via TLR7/8. Our study suggests that these cytokines in conditioned medium from RDgel-treated cells might contribute to apoptosis induction in colon26 cells.

We previously reported that DNA hydrogel containing an unmethylated cytosine-phosphate- guanine (CpG) DNA induced significant cytokine production and the delay of EG7-OVA tumor growth in vivo [14,22]. Cationized antigen could be efficiently loaded to DNA hydrogel and the antigen was released slowly [22,23]. Together with the previous results, the present findings suggest that loading of antigens, especially cationized ones, further increases the anti-tumor effect of the RDgel.

In conclusion, the RDgel developed in this study showed sustained RNA release and high immunostimulatory activity. RDgel can be formed by mixing two sets of self-assembled hexapodRD6. The results of the present study show that the RDgel can be used as an RNA adjuvant useful for cancer immunotherapy targeting TLR7/8.

## 4. Materials and Methods

### 4.1. Chemicals

RPMI 1640 medium was purchased from Nissui Pharmaceutical Co., Ltd. (Tokyo, Japan). Opti-modified Eagle’s medium (Opti-MEM) and fetal bovine serum (FBS) were obtained from Thermo Fisher Scientific Inc. (Waltham, MA, USA). Sodium chloride, sodium bicarbonate, potassium chloride, Tris-HCl, ethylenediaminetetraacetic acid (EDTA), ammonium chloride, and glucose were purchased from Wako Pure Chemicals Industries, Ltd. (Osaka, Japan). Monothioglycerol, MEM non-essential amino acids, and penicillin-streptomycin-glutamine mixed solution were purchased from Nacalai Tesque, Inc. (Kyoto, Japan). The 20 bp ladder was purchased from Takara Bio (Otsu, Japan). Propidium iodide, R848, and ovalbumin were obtained from Merck KGaA (Darmstadt, Germany). Alexa Fluor 488 annexin V/ Dead Cell Apoptosis kit with Alexa Fluor 488 annexin V and PI for Flow Cytometry was purchased from Invitrogen (San Diego, CA, USA). All other chemicals were of the highest grade available and were used without further purification.

### 4.2. Oligonucleotides

Phosphorothioate oligoribonucleotide (ORN)-1 and its labeled form with 6-FAM conjugated at the 3′-end (FAM-ORN-1) were purchased from FASMAC Corporation (Kanagawa, Japan). ORNs were synthesized at the 1 μmol scale and purified using HPLC. Phosphodiester oligodeoxynucleotides (ODNs) were purchased from Integrated DNA Technologies, Inc. (Skokie, IL, USA). ODNs were synthesized at the 100 nmol scale and desalted. ORN-1, ODN-1, ODN-2, ODN-3, ODN-4, ODN-5, and ODN-6 were designed to form hexapodRD6-1. ORN-1, ODN-7, ODN-8, ODN-9, ODN-10, ODN-11, and ODN-12 were designed to form hexapodRD6-2. The sequences of ORN-1 and the ODNs are listed in Table 1.

### 4.3. Cell Culture

RAW264.7 murine macrophage-like cells and colon26 murine colon carcinoma cells were grown in RPMI 1640 medium supplemented with 10% heat-inactivated FBS, 0.2% sodium bicarbonate, 100 IU/mL penicillin, 100 μg/mL streptomycin, and 2 mM l-glutamine at 37 °C in humidified air containing CO_2_. The murine dendritic cell line, DC2.4, was grown in RPMI 1640 medium supplemented with 10% heat-inactivated FBS, 0.2% sodium bicarbonate, 100 IU/mL penicillin, 100 μg/mL streptomycin, 2 mM l-glutamine, 50 μM monothioglycerol, and MEM non-essential amino acids at 37 °C in humidified air containing CO_2_.

### 4.4. Preparation of HexapodRD6 and RDgel

Each oligonucleotide was dissolved in Tris-EDTA buffer (10 mM Tris, 1 mM EDTA, pH 7.5). Appropriate molar ratios of the oligonucleotides included in 100–300 μM of hexapodRD6 were mixed with 150 mM sodium chloride. They were heated at 95 °C for 5 min to dissociate into single strands, 65 °C for 2 min, and 62 °C for 1 min to form double strand as designed, and were then slowly cooled to 4 °C at the rate of 2 °C per min. RDgel was prepared by mixing hexapodRD6-1 and hexapodRD6-2 at the room temperature. The formulation of the nanostructure and hydrogel was confirmed by PAGE. PAGE analysis was performed with a 10% polyacrylamide gel at 200 V for 60 min. Oligonucleotides were stained with SYBR GOLD and were observed using a LAS3000 imaging system (Fujifilm, Tokyo, Japan). FAM-labeled oligonucleotides were observed without SYBR GOLD staining. Separately, hexapodRD6 and RDgel were stained by mixing with 1 mg/mL of PI solution. Then, PI-labeled hexapodRD6 or RDgel were overlaid with PBS and their miscibility was evaluated.

### 4.5. RNA Release from RDgel

RNA release from RDgel was measured using a 12-well Transwell chamber system (#3401, pore size 0.4 μm, Corning, NY, USA). RDgel was dripped onto the Transwell and PBS was added to the lower well. Transwell was incubated at 37 °C and PBS was collected after 1, 3, 6, 12, and 24 h, and the absorbance from nucleic acid in PBS and fluorescence intensity derived from FAM-ORN-1 were measured. RNA release from RDgel with 50, 100, or 150 μM RNA/DNA was measured using the above method. RNA release from RDgel prepared with sodium ions (30, 150, 750 mM) or magnesium ions (2, 10, 50 mM) was also examined.

### 4.6. RNA Release from RDgel in 10% FBS

RDgel containing FAM-ORN-1 (FAM-RDgel) was overlaid with a solution containing nonheat-inactivated FBS at a final concentration of 10% and incubated at 37 °C. After 1, 2, 4, 6, 8, 16, or 24 h of incubation the supernatant was sampled and the fluorescence of the supernatant was measured using Wallac 1420 ARVO MX (Perkin Elmer, Waltham, MA, USA).

### 4.7. Cellular Uptake of RDgel by DC2.4 cells

DC2.4 cells on a 96-well plate at a density of 5 × 10^4^ cells/well were incubated with FAM-labeled ORN-1 or tetrapodRD3 diluted in 0.1 mL of Opti-MEM at a final concentration of 0.2 μM for 1, 2, 4, 8, and 16 h at 37 °C. Cells were harvested and washed twice with PBS. The fluorescent intensity of the cells was determined by flow cytometry (Beckman Courter, Indianapolis, IN, USA) using the Kaluza software (Beckman Courter, Indianapolis, IN, USA), and the mean fluorescence intensity (MFI) was calculated based on the histograms of the fluorescence intensity of the cells.

### 4.8. Cytokine Release from Immune Cells

DC2.4 cells and RAW264.7 cells were seeded on 96-well plates at a density of 5 × 10^4^ cells/well and incubated for 24 h before treatment. Then, ORN-1, hexapodRD6, or RDgel diluted in 0.1 mL of Opti-MEM at a final concentration of 1 μM were added to the cells. The supernatants were collected after 20 or 8 h and stored at –80 °C until use. The level of TNF-α in the supernatants was determined by ELISA using OptEIA sets (BD, San Diego, CA, USA).

### 4.9. Effects of Immune Cells Activated by RDgel on Tumor Cells

RAW264.7 cells seeded on 96-well plates at a density of 5 × 10^4^ cells/well were incubated with oligonucleotides (ORN-1, hexapodRD6, or RDgel) diluted in 0.1 mL of Opti-MEM for 8 h at 37 °C. The culture supernatant of RAW264.7 cells (culture medium) was added to colon26 cells seeded at a density of 5 × 10^4^ cells/well. The treated colon26 cells were harvested and the number of live cells were counted. The harvested cells were stained with Alexa Fluor 488-labelled anti-annexin V antibody and propidium iodide (PI) from the Alexa Fluor 488 annexin V/Dead Cell Apoptosis kit for Flow Cytometry. The proportion of the stained cells was determined by flow cytometry.

### 4.10. Statistical Analysis

The data were statistically evaluated by one-way analysis of variance followed by the Tukey–Kramer test for multiple comparisons and Student’s *t*-test for comparisons between two groups. A *p* value of < 0.05 was considered statistically significant.

## Figures and Tables

**Figure 1 molecules-25-00728-f001:**
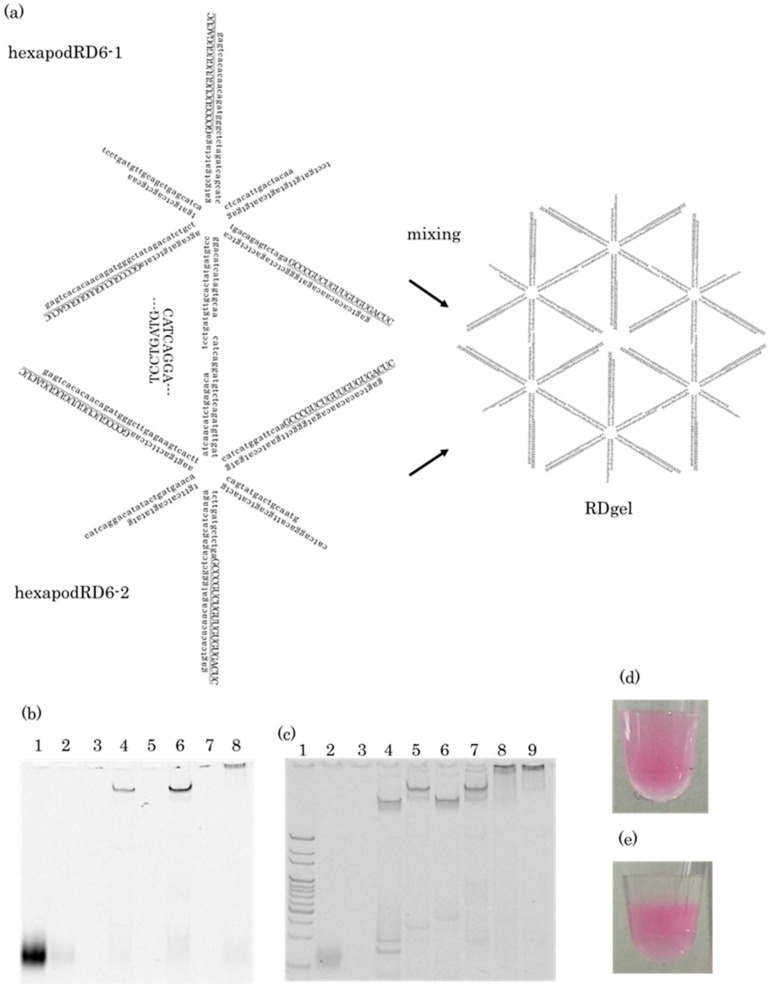
Preparation of hexapodRD6 and RDgel. (**a**) Putative structure of hexapodRD6 and the scheme for RDgel formation. The sequence of ORN-1 is underlined. Capital letters indicate phosphorothioate modification of inter-nucleotide linkages, and small letters indicate unmodified (phosphodiester) DNA. (**b**) Polyacrylamide gel electrophoresis (PAGE) analysis of FAM-ORN-1 and FAM-labeled nanostructured RNA/DNA based on FAM detection. Lane 1, FAM-ORN-1; lane 2, ORN-1; lane 3, hexapodD6-1; lane 4, FAM-hexapodRD6-1; lane 5, hexapodD6-2; lane 6, FAM-hexapodRD6-2; lane 7, hexapodD6-1 + hexapodD6-2; lane 8, FAM-hexapodRD6-1 + FAM-hexapodRD6-2. (**c**) PAGE analysis of ORN-1 and nanostructured RNA/DNA based on nucleic acid staining using SYBR GOLD. Lane 1, 20 base pair (bp) DNA ladder (TaKaRa Bio Inc., Otsu, Japan); lane 2, FAM-ORN-1; lane 3, ORN-1; lane 4, hexatetrapodD6-1; lane 5, FAM-hexapodRD6-1; lane 6, hexapodD6-2; lane 7, FAM-hexapodRD6-2; lane 8, hexapodD6-1 + hexapodD6-2; lane 9, FAM-hexapodRD6-1 + FAM-hexapodRD6-2. (**d**) Appearance of propidium iodide (PI)-stained hexapodRD6 after adding phosphate buffered saline (PBS). (**e**) Appearance of PI-stained RDgel after adding PBS.

**Figure 2 molecules-25-00728-f002:**
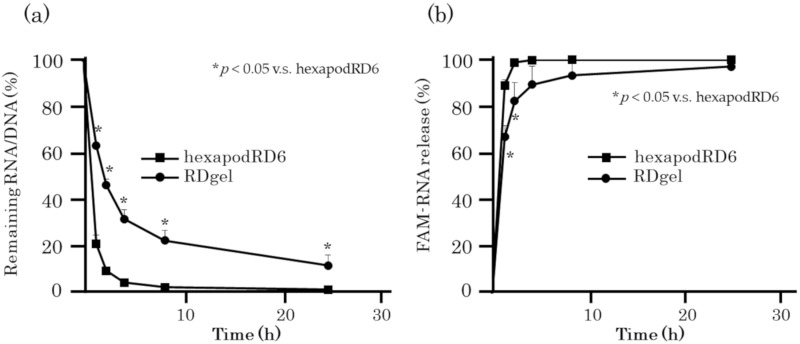
ORN-1 release from RDgel. (**a**) Time course of the amount of RNA/DNA remaining on the Transwell chamber after addition of 50 μM of RNA/DNA. The amount of RNA/DNA after 1, 3, 6, 12, and 24 h after the addition of RNA/DNA was calculated by subtracting the released amount from the total amount. (**b**) Time course of FAM-ORN-1 release from hexapodRD6 or RDgel. Fluorescence intensity of PBS in the lower chamber was measured at 1, 3, 6, 12, and 24 h after starting the experiment. The results are expressed as the mean ± SD of three independent experiments, * *p* < 0.05 compared with hexapodRD6.

**Figure 3 molecules-25-00728-f003:**
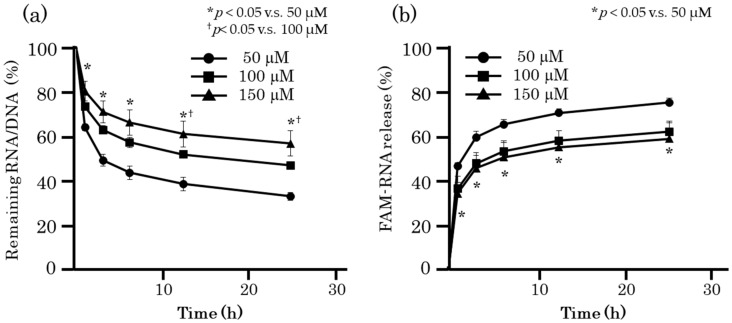
ORN-1 release from hexapodRD6 or RDgel prepared at different RNA/DNA concentrations. The same amount of 50, 100, or 150 μM RNA/DNA was added on the Transwell chamber. (**a**) Time course of the amount of RNA/DNA remaining on the Transwell chamber. The amount of RNA/DNA at 1, 3, 6, 12, and 24 h after addition was calculated by subtracting the released amount from the total amount. (**b**) Time course of FAM-ORN-1 released from the RNA/DNA hydrogel. The fluorescence intensity of PBS in the lower chamber was measured at 1, 3, 6, 12, and 24 h after starting the experiment. The results are expressed as the mean ± SD of three independent experiments, * *p* < 0.05 compared with 50 μM, ^†^
*p* < 0.05 compared with 100 μM.

**Figure 4 molecules-25-00728-f004:**
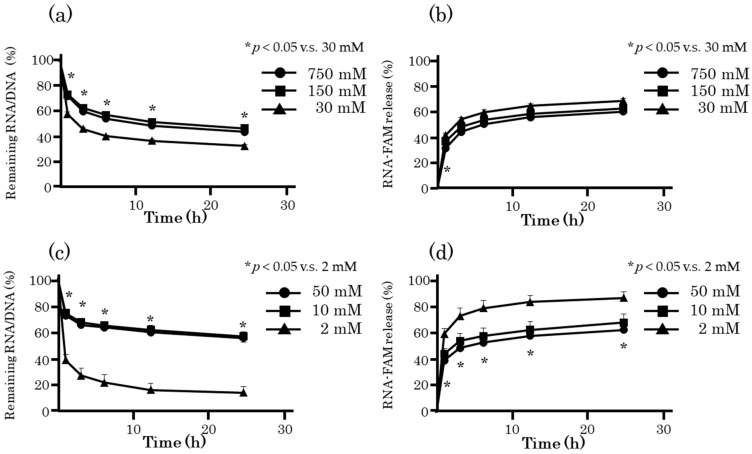
ORN-1 released from RDgel prepared with different metal ions. RDgel prepared with 30, 150, or 750 mM sodium ions, or with 2, 10, 50 mM magnesium ions was used. (**a**) Time course of the amount of RNA/DNA in RDgel prepared with sodium ions remaining on the Transwell chamber. The amount of RNA/DNA at 1, 3, 6, 12, and 24 h after RNA/DNA addition was calculated by subtracting the released amount from the total amount. (**b**) Time course of FAM-ORN-1 released from RDgel prepared with sodium ions. The fluorescence intensity of PBS in the lower chamber was measured at 1, 3, 6, 12, and 24 h after starting the experiment. (**c**) Time course of the amount of RNA/DNA in RDgel prepared with magnesium ions remaining on the Transwell chamber. The amount of RNA/DNA at 1, 3, 6, 12, and 24 h after RNA/DNA addition was calculated by subtracting the released amount from the total amount. (**d**) Time course of FAM-ORN-1 released from the RDgel prepared with magnesium ions. The fluorescence intensity of PBS in the lower chamber was measured at 1, 3, 6, 12, and 24 h after starting the experiment. The results are expressed as mean ± SD of three independent experiments, * *p* < 0.05 compared with 30 mM sodium ions or 2 mM magnesium ions.

**Figure 5 molecules-25-00728-f005:**
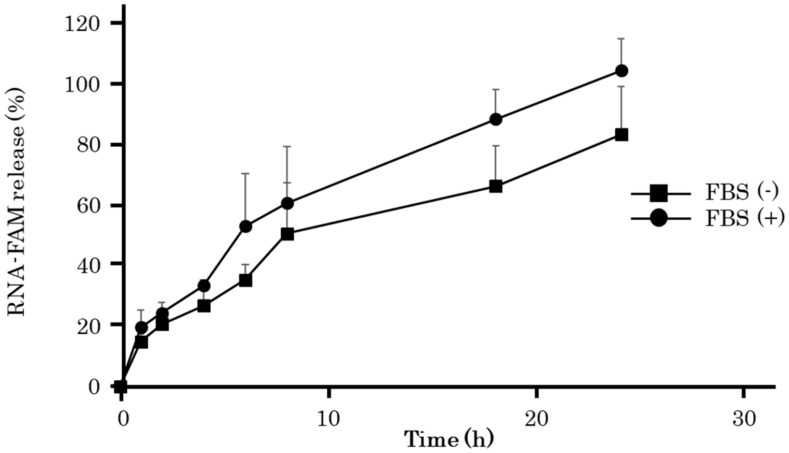
Time course of ORN-1 release from RDgel incubated with 10% FBS solution. The fluorescence intensity of the FBS solution was measured at 1, 2, 4, 6, 8, 16, or 24 h after starting the experiment. The results are expressed as the mean ± SD of three independent experiments.

**Figure 6 molecules-25-00728-f006:**
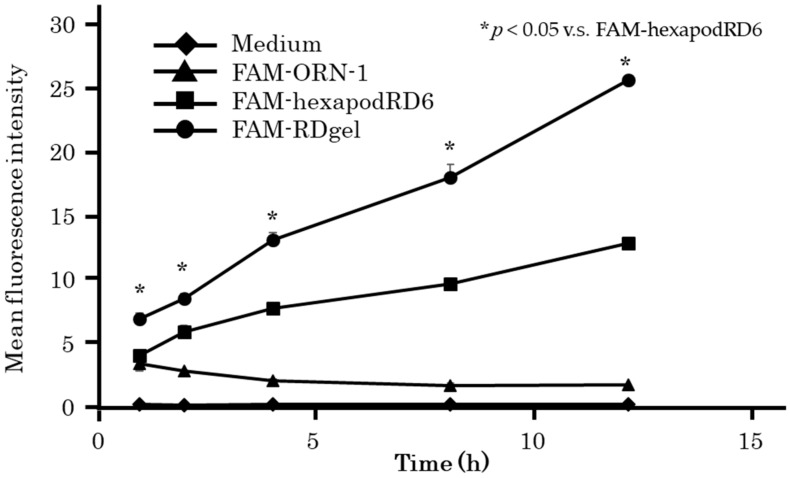
Uptake of ORN-1-FAM, FAM-hexapodRD6, and FAM-RDgel by DC2.4 cells. The mean fluorescence intensity of DC2.4 cells after addition of FAM-ORN-1, FAM-hexapodRD6, or FAM-RDgel was measured by flow cytometry. The results are expressed as mean ± SD of three independent experiments, * *p* < 0.05 compared with FAM-hexapodRD6.

**Figure 7 molecules-25-00728-f007:**
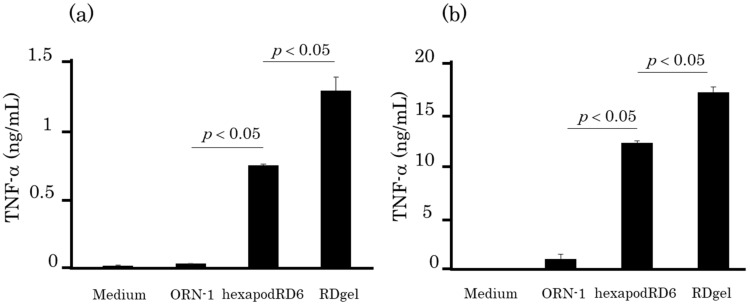
Tumor necrosis factor (TNF)-α release from immune cells after addition of ORN-1, hexapodRD6, and RDgel. TNF-α released from (**a**) DC2.4 cells or (**b**) RAW2.4 cells was measured at 20 h or 8 h, respectively, after RNA/DNA addition. The results are expressed as mean ± SD of three independent experiments.

**Figure 8 molecules-25-00728-f008:**
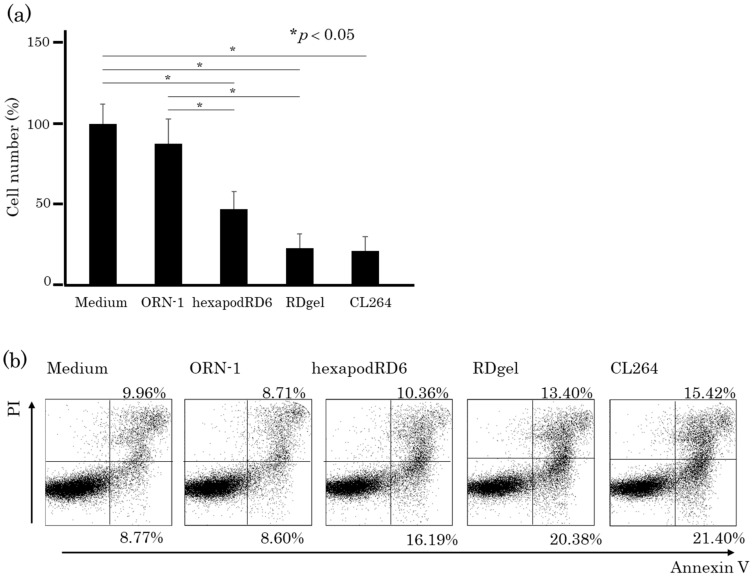
Effects of conditioned media of RAW264.7 cells on colon26 cancer cells. (**a**) Live colon26 cell counts after addition of conditioned medium. The conditioned medium was collected at 8 h after the addition of RNA/DNA. The results are expressed as mean ± SD of three independent experiments. (**b**) Apoptosis in colon26 cells after addition of conditioned medium. The proportion of colon26 cells stained with Alexa Fluor 488-labeled anti-annexin V antibody and propidium iodide (PI) was measured by flow cytometry.

**Table 1 molecules-25-00728-t001:** Sequences of oligonucleotides used to prepare hexapodRD6. Sequences in capital letters indicate phosphorothioate modification of the inter-nucleotide linkage. The asterisk (*) indicates the position of 6-FAM modification. All ODNs are phosphodiester linkages.

	Sequence (5′→3′)
ORN-1	GCCCGUCUGUUGUGUGACUC
FAM-ORN-1	GCCCGUCUGUUGUGUGACUC*
ODN-1	gagtcacacaacagatgggc tctagactctgtcaggacatcatagtgcaa
ODN-2	tcctgatg ttgcactatgatgtccagcagatgtctata
ODN-3	gagtcacacaacagatgggc tatagacatctgcttgatgctcagctgcaa
ODN-4	tcctgatg ttgcagctgagcatcagatgctgatctaga
ODN-5	gagtcacacaacagatgggc tctagatcagcatcctcacattgactacaa
ODN-6	tcctgatg ttgtagtcaatgtgagtgacagagtctaga
ODN-7	gagtcacacaacagatgggcttgaatccatgatgcagtatgactgcaatg
ODN-8	catcagga cattgcagtcatactgtcttgatgctctga
ODN-9	gagtcacacaacagatgggctcagagcatcaagatgttcatcagtatatg
ODN-10	catcagga catatactgatgaacaaagtgacttctcaa
ODN-11	gagtcacacaacagatgggcttgagaagtcacttatcaacatctgagaca
ODN-12	catcagga tgtctcagatgttgatcatcatggattcaa

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
