# Peer review of "Development of RNA/DNA Hydrogel Targeting Toll-Like Receptor 7/8 for Sustained RNA Release and Potent Immune Activation"

_molecules, 2020, doi:10.3390/molecules25030728_

Round 1

Reviewer 1 Report

The manuscript by Komura at co-workers describe the generation of a novel DNA/RNA hydrogel which targets the toll-like receptors 7 and 8 in vitro. A therapeutic RNA strand is included within the hydrogel sequence which binds to the toll-like receptors. The uptake of this fragment from the hydrogel is quantified in serum containing solutions towards immune cells DC2.4 and the subsequent release of TNF-a with cancer-cell killing properties. This new approach to target the TNL receptors should generate impact in the immediate and wider community. I support the publication of this well written manuscript.

Minor comments:

Figure 1A is not clear:

DNA and RNA sequences are added, but information about 5’ and 3’ termini and how the strands are connected along the middle is missing. Please provide separate diagrams of hexapod RD6-1 and -2, and its conversion to RDgel to aid clarity. The caption does not define the small letters.

Some relevant confocal laser scanning microscopic images in figure 6 would boost the visual representation of these promising results.

Line 110, please add a definition of the Transwell chamber system and the expected results, plus the cavity size.

Section 2.4 please add a comment why the RNA sequence is released.

Methods section:

Important missing technical information about hydrogel formation, annealing concentration, annealing rate not complete. Figure 1 d and e mentions PI stain, but missing technical information about this experiment. DNA and RNA synthesis scale and purification missing. Immune cells missing hydrogel concentration.

Reviewer 2 Report

Manuscript concerns the preparation of hydrogels based on DNA and guanosine- and uridine-rich single-stranded RNA (GU-rich RNA). Such structures were considered for sustained release of GU-rich RNA that induces strong immune response essential in cancer treatment. Paper is well-constructed. Applied methodology has ben selected properly, obtained results have been discussed in an adequate manner. Manuscript is definitely worth notocing. However, some minor revisions are needed.

In section Introduction Authors mentioned that: ”(…) various types of sustained release systems for bioactive RNAs have been reported”. Some of them need to be listed and briefly characterized. In section 3. Authors mentioned that it was proved during their previous research that DNA hydrogel containing CpG DNA induced significant cytokine production and anti-tumor effects in vivo. Some more sentences explaining these anti-tumor effects should be added. In section concerning „Preparation of hexapodRD6 and RDgel” Authors described the following reaction route: „(…) they were heated at 95°C for 5 min, 65°C for 2 min, and 62°C for 1 min; and were then slowly cooled to 4°C”. Brief description explaining why exactly such a procedure was applied is necessary. Section 4.7.: the methodology of calculation of the mean fluorescence intensity should be presented in more detail. At what temperature the RNA release from RDgel was determined? There is no information concerning this issue.
